# EGFR Tyrosine Kinase Inhibitor Efficacy in Older Adult Patients with Advanced EGFR-Mutated Non-Small-Cell Lung Cancer: A Meta-Analysis and Systematic Review

**DOI:** 10.3390/medicina58111645

**Published:** 2022-11-15

**Authors:** Chang-Hung Chen, Deng-Wei Chou, Kuo-Mou Chung, Han-Yu Chang

**Affiliations:** Department of Chest Medicine, Tainan Municipal Hospital (Managed by Show Chwan Medical Care Corporation), Tainan 70133, Taiwan; choudw@gmail.com (D.-W.C.); 2l0642@tmh.org.tw (K.-M.C.); dr.xlung@gmail.com (H.-Y.C.)

**Keywords:** EGFR tyrosine kinase inhibitors, older adult patients, non-small cell lung cancer

## Abstract

*Background and Objectives*: Lung cancer remains the most common malignancy worldwide. As the global population ages, the prevalence of epidermal growth factor receptor (EGFR)-mutation-positive non-small cell lung cancer (NSCLC) is increasing. *Materials and Methods*: We performed a meta-analysis and a systematic review of randomized, controlled trials to evaluate the efficacy of EGFR TKIs on progression-free survival (PFS) and overall survival (OS) in older adult patients with advanced EGFR-mutated NSCLC. A total of 1327 patients were included; among these, 662 patients were >65 years of age. *Results*: A pooled analysis indicated (1) an overall improvement in higher PFS for dacomitinib and osimetinib than that for other drugs (hazard ratio [HR] = 0.654, 95% CI: 0.474 to 0.903; *p* = 0.01) and (2) and no significant difference in the OS between the EGFR TKIs (HR = 0.989, 95% CI: 0.796 to 1.229; *p* = 921). *Conclusion*: Our study found that osimertinib achieved a higher PFS than all other EGFR TKIs did. Osimertinib is the preferred EGFR TKI for treatment of older adult patients with advanced EGFR-mutated NSCLC.

## 1. Introduction

Lung cancer is the leading cause of cancer-related death worldwide, and the majority of diagnosed lung cancer patients have non-small-cell lung cancer (NSCLC). About 70% of patients with NSCLC are diagnosed at an advanced stage, and the median age at diagnosis is 70 years [1,2]. Conventionally, “elderly” or “older adult” has been defined as a chronological age of 65 years old or older, while those patients who are 65 through to 74 years old are referred to as “early elderly”, and those over 75 years old are “late elderly” [3]. Age is an important factor in making management decisions because of the complex interplay between normal age-related decline and comorbidities [1]. Clinicians who treat older adult patients with lung cancer must consider age-related factors, such as poor functional status, a high comorbidity burden, and polypharmacy when making treatment decisions [4]. Over the past few decades, the identification of oncogenic drivers predicting the clinical response to targeted therapies produced a radical shift from the histological to the molecular subtyping of lung cancer, thus establishing a new paradigm of precision medicine [5]. The epidermal growth factor receptor (EGFR) gene-activating mutations represented the first molecular predictive biomarker which was discovered in lung cancer in 2004, underlying clinical responsiveness to the EGFR tyrosine kinase inhibitors (TKIs) [6]. In the case of epidermal growth factor receptor (EGFR) mutation-positive non-small-cell lung cancer (NSCLC), age-related factors might be less influential because the first-line treatment generally comprises a monotherapy with EGFR tyrosine kinase inhibitors (TKIs). However, although these agents are generally associated with manageable tolerability profiles, no specific recommendations are available for the use of EGFR TKIs on the basis of the patient’s age [7]. The results of a meta-analysis conducted by Roviello et al. revealed that EGFR TKIs significantly slow disease progression and are effective as a preferred clinical treatment option for older adults [8]. Wu et al. reported that there was no statistically significant progression-free survival (PFS) and overall survival (OS) in the patients > 75 years old sub-group in the head-to-head LUX-lung 7 trial that compared the efficacy of afatinib and gefitinb [9]. Wu et al. did not mention the result of the progression-free survival (PFS) and overall survival (OS) measures in the group with between 65 and 74-year-old patients [8]. So the data on the impact of EGFR TKIs on progression-free survival (PFS) and overall survival (OS) in patients > 65 years old are not conclusive. In recent years, several multi-center, international, exploratory clinical, randomized, controlled trials (RCTs) have compared various the EGFR TKIs. Thus, the present study performed a systematic review and a meta-analysis of the available clinical data from representative head-to-head EGFR TKIs comparison RCTs to evaluate the efficacy of the EGFR TKIs on the progression-free survival (PFS) and overall survival (OS) of older adult patients with advanced EGFR-mutated NSCLC.

## 2. Materials and Methods

### 2.1. Data Retrieval Strategies

We conducted a meta-analysis of RCTs in accordance with the PRISMA Statement [10]. The relevant publications were identified from PubMed, EMBASE, MEDLINE, and the Cochrane Library using the following search terms: ‘‘non-small cell lung cancer”, “older adult”, “elderly patient”, “EGFR TKIs’’, “erlotinib’’, “gefitinib’’, “afatinib’’, “osimertinib” and “dacomitinib”. The search was performed without any restrictions on the country or article type. Only English language publications were included. The reference lists of all of the selected articles were independently screened to identify additional studies that had been left out of the initial search.

The publications available in these databases up to 31 January 2022 were analyzed. The search criteria were limited to articles reporting phase II or phase III RCTs. The computer search was supplemented with a manual search of the primary studies that were referenced in all of the retrieved review articles.

### 2.2. Inclusion Criteria

Two authors screened the studies according to specific inclusion and exclusion criteria. In the event of a disagreement, a third author was involved in the decision-making process. Older adult patients were defined as patients that were >65 years old. The studies to include in our analysis were identified according to the following criteria: (1) participants with advanced EGFR-mutated NSCLC; (2) having had an anti-EGFR TKI- based therapy; (3) the presence of a control arm for comparison (other anti-EGFR therapy), (4) OS and PFS being expressed as the hazard ratio (HR). The following were the exclusion criteria: (1) insufficient data being available to estimate the outcomes; (2) animal studies; (3) sample size of each arm < 10 participants; (4) non-randomized studies.

### 2.3. Data Extraction and Quality Assessment

Two authors independently extracted the relevant data including the name of the first author, the publication year, the drug that was administered, the patient demographics, the study design, the PFS HR, and the OS HR. We assessed the study quality of the evidence based on the RoB 2: a revised tool for assessing the risk of bias in randomized trials [11]. The protocol for this systematic review was registered on PROSPERO (CRD42022322091) and is available in full on the website at http://www.crd.york.ac.uk/PROSPERO (accessed on 10 May 2022).

### 2.4. Statistical Analysis

The statistical analyses were performed using CMA software version 3.3.070. The summary estimates were generated using a fixed effects model (the Mantel–Haenszel method) or a random effects model (the DerSimonian–Laird method); the model used depended on whether heterogeneity was present. The statistical heterogeneity was assessed using the Q test and the I^2^ statistic. I^2^ values of 25%, 50%, and 75% were considered to indicate low, moderate, and high heterogeneity, respectively. For the PFS and OS, the HRs with 95% confidence intervals (CI) were calculated for each study. For all of the statistical analyses, a value of *p* < 0.05 was regarded as statistically significant, and all of tests were two-sided.

## 3. Results

Our search yielded 3337 potentially relevant articles. A total of 2883 studies were excluded as duplicates. After viewing the titles and abstracts of the remaining four hundred and fifty-four studies, the full texts of twenty-seven studies were retrieved, and six studies [12,13,14,15,16,17] were included in the analysis (Figure 1).

The characteristics of the studies are summarized in Table 1. Two phase III studies and one phase IIb study were identified (Table 1). These studies compared pairs of EGFR TKIs: afatinib with gefitinib [12], dacomitinib with gefitinib [14], and osimertinib with gefitinib and erlotinib [16]. A total of 1327 patients were included, among whom 579 patients were >65 years of age. In the LUX-Lung 7 trial, there was no significant PFS and OS benefit when they were comparing afatinib with gefitinib in older adult participants [12,13]. In the ARCHER 1050 trial, there was significant PFS benefit when they were comparing dacomitinib with gefitinb (HR 0.69, 0.48–0.99) in older adult participants, but there was no OS benefit [14,15]. In the FLAURA trial, there was significant PFS benefit when they were comparing osimertinib with erlotinib and gefitinib (HR 0.49, 0.35–0.67) in older adult participant, but there was no OS benefit [16,17].

We assessed the quality of the evidence of these selected RCTs based on the revised tool for assessing the risk of bias in randomized trials (RoB 2) (Figure 2). These six studies provided the adequate study design and good data quality for our analysis, and the overall risk of bias was low.

A pooled analysis revealed an overall improvement in the PFS for older adult patients with advanced EGFR-mutated NSCLC who were treated with dacomitinib and osimertinib (HR = 0.654, 95% CI: 0.474 to 0.903; *p* < 0.01; Figure 3). A random effects model was used for the analysis of the PFS due to the presence of moderate heterogeneity (I^2^ = 60.8%) among the trials. The pooled analysis revealed no significant difference in the OS for older adult patients with advanced EGFR-mutated NSCLC who were treated with different EGFR TKIs (gefitinib, erlotinib, afatinib, dacomitinib and osimertinib) (HR = 0.989, 95% CI: 0.796 to 1.229; *p* = 0.921; Figure 4). The fixed effects model was used for the analysis of the OS due to the lack of heterogeneity (I^2^ = 0%) among the trials.

## 4. Discussion

EGFR TKIs such as erlotinib, gefitinib, afatinib, osimertinib, and dacomitinib are active agents that are used to treat patients with advanced EGFR-mutated NSCLC. Compared with cytotoxic chemotherapy, EGFR TKIs increase the tumor response rate to a greater degree, further prolong the PFS, cause fewer adverse effects, and improve the health-related quality of life. Few studies have demonstrated that EGFR TKIs result in a higher OS than standard chemotherapy does, but the majority of trials allowed the participants to switch treatments upon the disease’s progression, which has a confounding effect on any OS analysis [18]. Because the aforementioned EGFR TKIs are therapeutically superior to the cytotoxic chemotherapy, the current standard of first-line care for patients with advanced EGFR-mutated NSCLC is the treatment with EGFR TKIs. With the aging of the global population, the prevalence of advanced EGFR-mutated NSCLC is increasing. Roviello et al. found that EGFR TKIs significantly slow the disease’s progression and are effective as a clinical treatment option for older adult patients with advanced EGFR-mutated NSCLC [9]. More research is required to determine which EGFR TKI is the most suitable for older adult patients with advanced EGFR-mutated NSCLC.

In the CTONG0901 trial, PFS and OS did not significantly differ between the gefitinib and erlotinib groups [19]. In the LUX-Lung 7 trial, PFS and OS did not significantly differ between the gefitinib and afatinib groups [12,13], however, a subgroup analysis by age reported that there was a higher favored PFS, but not an OS in the afatinib group than there was in the gefitinib group for the patients > 75 years old [8]. The ARCHER 1050 trial reported significantly higher PFS and OS in the dacomitinib group than there was in the gefitinib group [14,15]. The FLAURA trial reported significantly higher PFS and OS in the osimertinib group than there was in the erlotinib and gefitinib groups [16,17]. Lin et al. found that out of the EGFR TKIs they tested, osimertinib achieved the highest PFS; the results for the OS were unavailable due to there being incomplete data [20]. A meta-analysis by Holleman et al. indicated that among all of the EGFR TKIs, osimertinib achieved the highest PFS and OS values, and that gefitinib, erlotinib, and osimertinib were associated with the fewest toxicities [21]. Zhao et al. reported that (1) osimertinib monotherapy achieved higher PFS than that which was achieved with dacomitinib, afatinib, erlotinib, and gefitinib, and (2) osimertinib, and gefitinib plus a pemetrexed-based chemotherapy achieved the highest OS [22]. Studies have consistently shown that EGFR TKIs are well tolerated, however, adverse events including those of the skin, gastrointestinal tract, and lungs are commonly seen [23]. These side effects are mild in most cases, but they can affect the quality of a patient’s life and subsequently lead to dose reduction and treatment discontinuation. Individual EGFR TKIs were associated with different toxicity spectrums. The generally greater toxicity of dacomitinib and afatinib have been noted in previous studies [24].

Our pooled analysis indicated an overall improvement in the PFS that was treated with dacomitinib and osimetinib and no significant difference in the OS that was treated with different EGFR TKIs. A meta-analysis by Holleman et al. revealed that osimertinib had a more significant PFS benefit than other EGFR TKIs did [21]. However, this analysis did not show a formal statistical significance for the OS because the final OS data of FLAURA trial were not available during their study period. Our analysis concentrates on the PFS and OS benefits in older adult patients. The final OS data were completely collected. The PFS efficacy of osimertinib in our analysis is consistent with previous studies, but the OS benefit does not show any significant difference. We consider the presence of multiple comorbidities to offset the OS benefits of dacomitinib and osimertinib. Furthermore, the combination of EGFR TKIs and cytotoxic agents increases the likelihood of adverse events and is, thus, not a practical therapeutic strategy for older adult patients. In the latest national comprehensive cancer network (NCCN) guidelines of non-small cell lung cancer, gefitinib, erlotinib, afatinib, dacomitinib, and osimertinib are all category 1 recommended EGFR TKIs [25]. The gefitinib option, which was launched in 2000, and still plays an important role in the lung cancer treatment. It has not been removed from current clinical practice yet. The greater toxicity of afatinib and dacomitinib should be considered in older adult lung cancer treatment. To the best of our knowledge, the present study is the most up to date one in the literature. It was conducted based on data that were extrapolated from trials conducting head-to-head comparisons of EGFR TKIs. Our data covered a total of 1372 patients and evaluated the clinical effect of EGFR TKIs in the treatment of older adult patients with advanced EGFR-mutated NSCLC. The limitation of our study is that we excluded the retrospective and cohort studies. A real-world retrospective data analysis would probably reveal different result. In conclusion, osimertinib has the highest therapeutic efficacy, causes the fewest adverse effects, and is, therefore, the best EGFR TKI for older adult patients with advanced EGFR-mutated NSCLC.

## 5. Conclusions

Our study demonstrated that osimertinib achieved a higher PFS than all of the other EGFR TKIs did in older adult patients with advanced EGFR-mutated NSCLC. Furthermore, we found that dacomitinib and afatinib have generally greater toxicity than the other EGFR TKIs do. Therefore, we believe that osimertinib is the best EGFR TKI for older adult patients with advanced EGFR-mutated NSCLC.

## Figures and Tables

**Figure 1 medicina-58-01645-f001:**
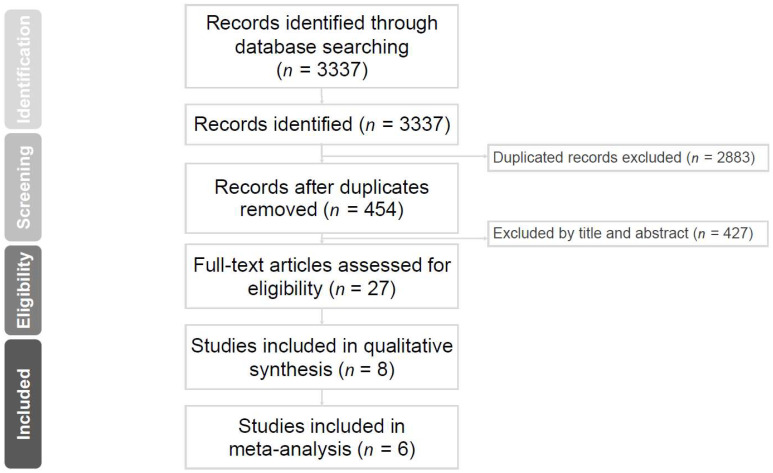
Flow diagram for study selection.

**Figure 2 medicina-58-01645-f002:**
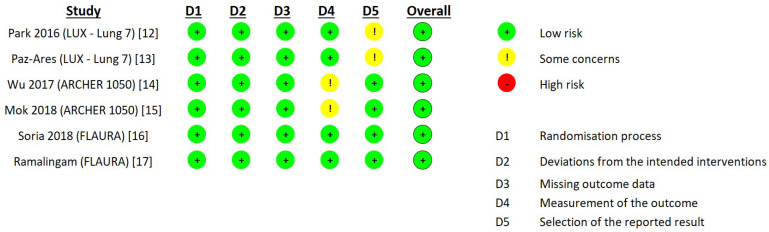
Risk of bias summary: review authors’ judgments about each risk of bias item for each included study.

**Figure 3 medicina-58-01645-f003:**
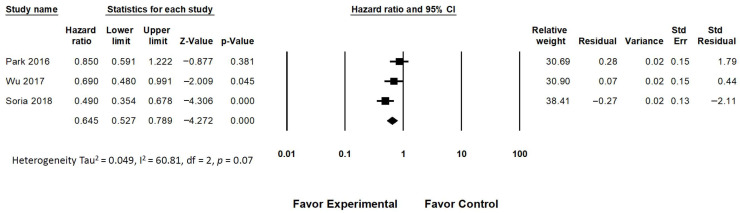
Analysis for the progression-free survival of the EGFR TKIs in older adult patients (The effect size for each study is represented by a square, with the location of the square representing both the direction and magnitude of the effect. The summary effect is represented by a diamond. The location of the diamond represents the effect size while its width reflects the precision of the estimate.) [12,14,16].

**Figure 4 medicina-58-01645-f004:**
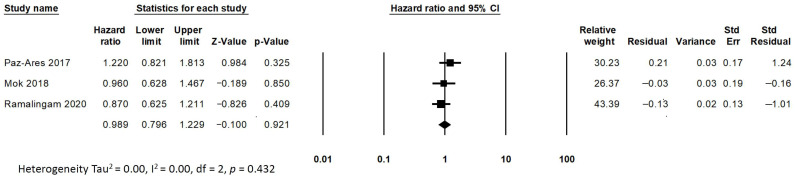
Analysis for the overall survival of the EGFR TKIs in older adult patients (The effect size for each study is represented by a square, with the location of the square representing both the direction and magnitude of the effect. The summary effect is represented by a diamond. The location of the diamond represents the effect size while its width reflects the precision of the estimate.) [13,15,17].

**Table 1 medicina-58-01645-t001:** Characteristics of the analyzed trials.

Study Authors Year	Phase	Treatment Experimental/Control (EGFR TKIs) Participants	Old Adult Participants Old Adults (>65 Years Old)/Total	Primary Endpoint PFS and OS 95% CI
**LUX-Lung 7** Park K. et al. 2016 [12] Paz-Ares L. et al. 2017 [13]	IIB	Afatinib/Gefitinib 160/159	142/319 (44%)	**PFS** (Total/Old adults) HR 0.73 (0.57–0.95)/HR 0.85 (0.59–1.22) **OS** (Total/Old adults) HR 0.86 (0.66–1.12)/HR 1.22 (0.82–1.81)
**ARCHER 1050** Wu YL. et al. 2017 [14] Mok TS. et al. 2018 [15]	III	Dacomitinib/Gefitinib 227/225	179/452 (39%)	**PFS** (Total/Old adults) HR 0.59 (0.47–0.74)/HR 0.69 (0.48–0.99) **OS** (Total/Old adults) HR 0.76 (0.58–0.99)/HR 0.96 (0.63–1.47)
**FLAURA** Soria JC. et al. 2018 [16] Ramalingam SS. et al. 2020 [17]	III	Osimertinib/Erlotinib & Gefitinib 279/277	258/556 (46%)	**PFS** (Total/Old adults) HR 0.46 (0.37–0.57) / HR 0.49 (0.35–0.67) **OS** (Total/Old adults) HR 0.80 (0.64–1.00)/HR 0.87 (0.63–1.22)

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
