# Peer review of "EGFR Tyrosine Kinase Inhibitor Efficacy in Older Adult Patients with Advanced EGFR-Mutated Non-Small-Cell Lung Cancer: A Meta-Analysis and Systematic Review"

_medicina, 2022, doi:10.3390/medicina58111645_

Round 1

Reviewer 1 Report (Previous Reviewer 2)

The article 'EGFR Tyrosine Kinase Inhibitor Efficacy in Older Adult Patients With Advanced EGFR Mutated Non-Small Cell Lung Cancer: A Meta-Analysis and Systematic Review', submitted by Chen CH et al., finds osimertinib exhibits higher progression-free survival (PFS) compared to other EGFR-TKIs for the elderly patients with advanced EGFR-mutated non-small-cell lung cancer (NSCLC). The authors have performed meta-analysis and systematic review of the available clinical data of randomized controlled trials (RCTs) to evaluate the efficacy of EGFR-TKIs on PFS in older adult patients with advanced EGFR-mutated NSCLC. This study could be useful for future therapy planning for EGFR-mutated NSCLC elderly patients. I have a few concerns here-

1.     Authors should improve the introduction section of the article, it is not sufficient for this study.

2.     Instead of just saying that the charractristics of the study have been shown in table 1, the authors should explain their findings.

3.     Fig 3 and Fig 4 are not explained adequately.

4.     Correct the spelling mistakes, for eg. re-vised

Author Response

Reviewer 2 Report (New Reviewer)

This is a great met analysis looking at the data of EGFR TKI in older population in terms of Progression free survival and overall survival

Patient with Osimertinib has better progression free survival. No difference in the different TKI's in terms of OS. 

Would like to see a pooled demographic and comorbidities data from these studies to look for clinical indicators of any racial and gender differences.

Author Response

Reviewer 3 Report (New Reviewer)

Reviewer found more sufficient systematic review and meta-analysis for related articles was published in the high-impact journals. And reviewer does not find any more emerging decision and issues. Lack of research strategies and analysis.

https://bmccancer.biomedcentral.com/articles/10.1186/s12885-022-09444-0

https://www.sciencedirect.com/science/article/pii/S2352047722000077

https://www.ncbi.nlm.nih.gov/pmc/articles/PMC9186167/

Round 2

Reviewer 1 Report (Previous Reviewer 2)

Authors have addressed my suggestions.

Reviewer 3 Report (New Reviewer)

Dear Authors:

Revised manuscript much better compare submitted manuscript; however, the reference list follows the journal format.

This manuscript is a resubmission of an earlier submission. The following is a list of the peer review reports and author responses from that submission.

Round 1

Reviewer 1 Report

The authors performed an interesting meta-analysis of clinical trials on EGFR mutation-positive lung adenocarcinoma.

However, it is difficult to evaluate a meta-analysis when only six of the 3300 data were amenable to meta-analysis, and a close reading of the individual papers would be more beneficial. In particular, for the treatment of EGFR-positive lung cancer, I believe that osimertinib and afatinib are currently mainly used in the first line, and I do not see the value of a meta-analysis in this setting, since a direct comparison of these two drugs is not possible.

Reviewer 2 Report

The article 'EGFR Tyrosine Kinase Inhibitor Efficacy in Older Adult Patients With Advanced EGFR Mutated Non-Small Cell Lung Cancer: A Meta-Analysis and Systematic Review', submitted by Chen CH et al., finds osimertinib exhibits higher progression-free survival (PFS) compared to other EGFR-TKIs for the elderly patients with advanced EGFR-mutated non-small-cell lung cancer (NSCLC). The authors have performed meta-analysis and systematic review of the available clinical data of randomized controlled trials (RCTs) to evaluate the efficacy of EGFR-TKIs on PFS in older adult patients with advanced EGFR-mutated NSCLC. This study could be useful for future treatment planning for EGFR-mutated NSCLC elderly patients. I have a few concerns here-

1.     Authors should improve the introduction section of the article, it is not sufficient for this study.

2.     Authors need to elaborate the results section. Table-1 content is hardly explained like- Lux Lung-7, ARCHER 1050, and FLAURA studies. Data presentation is poor; make proper figures as these are difficult to comprehend.

3.     Authors must discuss how their findings are different from the study conducted by Holleman et al. (Ref 17).

Reviewer 3 Report

The manuscript by Chen et al, entitled "EGFR Tyrosine Kinase Inhibitor Efficacy in Older Adult Patients With Advanced EGFR Mutated Non-Small Cell Lung Cancer: A Meta-Analysis and Systematic Review" is written in good English but a wording/spelling work-over is needed.

The introduction covers necessary information.

In M&M, the search items are mostly reasonable, but the item "older adults" limits the result in higher extent unless it seems that the authors did not include studies who might name it "older patients" or general studies stratifying older patients in a separate group.

As authors state themselves that anti-EGFR therapy in NSCLC is often performed and reflected in many studies, they included only 6 studies what seems underrepresenting the research status sufficiently.

In results, studies were analyzed concerning survival.

In the discussion results are put in relation to present published data adequately.

Next to language revision, I can recommend the manuscript for publication unless editors should be aware of the mentioned limitations.

Round 2

Reviewer 2 Report

Data in Fig 3 is duplicated and authors have not responded clearly how their study is different from that of Holleman et al. 
